# Soil Carbon and Nitrogen Forms and Their Relationship with Nitrogen Availability Affected by Cover Crop Species and Nitrogen Fertilizer Doses



**Lucas Boscov Braos** [1,2,*], **Roberta Souto Carlos** [1], **Aline Carla Trombeta Bettiol** [1], **Marina Ali Mere Bergamasco** [1], **Maira Caroline Terçariol** [1], **Manoel Evaristo Ferreira** [1] **and Mara Cristina Pessôa da Cruz** [1]

1   School of Agricultural and Veterinarian Sciences, São Paulo State University (Unesp), Jaboticabal 14884-900, São Paulo, Brazil
2   Instituto Federal de Educação, Ciência e Tecnologia do Sul de Minas Gerais, Campus Inconfidentes, Inconfidentes 37576-000, Minas Gerais, Brazil
*   Correspondence: lucasbraos@hotmail.com; Tel.: +55-16-997504659

**Abstract:** Cover crops and N fertilization strongly impact the forms of soil organic C and N and their availability, which change the responses of plants to N fertilization and soil organic C accumulation. Our study objectives were to evaluate the effects of cover crops and N doses on soil total and soluble C and N contents, N fractions, and potentially available N in a long-term no-till experiment. The experiment was conducted in a randomized block design with split plots and four replicates. The main treatments were cover crops species, jack bean, lablab bean, millet, velvet bean, and fallow cultivated prior to maize. Secondary treatments included two doses of mineral N (0 and 120 kg ha$^{-1}$). Soil samples were collected at depths of 0–5, 5–10, 10–20, and 20–40 cm, which were analyzed for total and water-soluble C and N contents, N fractions (acid hydrolysis method), and potentially available N (hot KCl solution and direct steam distillation methods). Cover crop velvet bean resulted in the highest soil organic carbon levels, and cover crop millet plus fertilization resulted in the highest levels of soil total N. The amino sugar was the largest N fraction, which decreased by 8% with N fertilization. The soluble C and N content strongly correlated with total and available N content. The changes in soil N were influenced by cover crop species and fertilization and the interactions of both, so the combination of fertilization regime and cover crops must be chosen with care. Additionally, legumes are a good source of plant and soil N in systems with low input of N via fertilizer; however, the combination of N fertilizer with legumes can reduce soil N reserves, leading to its long-term depletion.

**Keywords:** green manure; winter crops; no tillage; N fraction; potentially available N; water soluble C

## 1. Introduction

Agricultural methods were intensified in recent decades to increase food production, but the methods used have led to issues such as soil degradation. To meet the demand for food and minimize the risks of soil degradation, farmers worldwide are increasingly adopting the no-till system [1]. This system helps to maintain the stocks of organic carbon (C) in the soil and reduces erosion. The adoption of the no-tillage system results in less soil disturbance and increased input of plant biomass, consequently increasing the content of organic matter, which contributes to increasing contents of C and nitrogen (N) in the soil. This includes labile forms, which, consequently, may increase their availability N [2–4]. These changes are related to biomass input, decreased soil disturbance, increased soil aggregation, crop rotation, etc. The system has some limitations, such as stratification of the nutrients in the soil profile, superficial compaction, and acidification of deeper layers. However, its adoption has been preferred to the conventional soil tillage system in Brazil and many other countries since the 2000 s [1,5,6].

Cover crops are one of the largest sources of biomass for the soil under this management system and, therefore, also interfere with soil organic C and N dynamics. These crops are largely used to cover soil during dry seasons and can include green manure and alternative plants in crop rotations. Each cover crop has a different effect on soil C and N dynamics; the volume produced and the crop residue composition are highly variable [3,7]. This heterogeneity affects C and N mineralization and immobilization processes; changes N availability, efficiency of N fertilizer, and N uptake by plants; and influences their movement to deeper soil layers, resulting in changes in crop productivity [4,8,9]. These effects are even more notable in places under successive surface applications of these plant residues in the long term [10]. In addition, N fertilization affects the C and N dynamics in the soil. Recently, the possibility that N fertilization can reduce the soil organic C and total N contents has been discussed [11,12], although the positive effects on soil C and N contents are often assigned to N fertilization [3].

The interaction between N fertilization and the type of cover crop strongly influences the soil organic carbon (SOC) dynamics because fertilization affects the production of plant biomass, microbial activity, and the availability and uptake of N by plants and microorganisms. This is related to N affecting both the productivity of these crops and the decomposition and subsequent transformations of the biomass. The accumulation of SOC in soils cultivated with grasses or crops with a high C:N ratio are highly dependent on the addition of N via fertilizer [3,9,13]. Although SOC and soil total nitrogen (STN) are important measures in the assessment of soil management systems, they may not reflect changes in the lability and dynamics of these elements in the soil, mainly because the nitrogenous organic substances in the soil are not readily available. Thus, the STN content is rarely used as a measure of available N [14]. The water-extractable organic C (WEOC) and N (WEON) are the most labile and biodegradable forms of soil organic matter and play an important role in the cycle of these elements and the supply of N to plants, in addition to being more sensitive to changes in soil management [2,15]. Thus, the evaluation of WEOC and WEON can be more suitable for determining the effects of a management system on N dynamics and N availability for crops.

Another method of assessing the availability and transformations of organic N induced by different management practices is the application of the chemical fractionation of organic N or the single extraction of potentially available N [14,16,17]. Fractionation allows the separation of soil N into different pools, each with different environmental and agronomic importance, with the forms associated with amino sugars and amino acids considered the most important reservoirs of available N given their ability to be quickly available to plants. Identifying the transformation of soil N fractions may help to understand the fate of N in different cropping systems, enabling the identification of those systems that result in lower losses and increased accumulation and availability of N for plants. However, fractionation schemes are unsuitable for routinely measuring N availability. Potentially available N estimation methods can be used to extract specific fractions of soil organic N that are made available during the crop cycle. The results of such methods are correlated with the N absorbed by plants, and can support the N fertilizer recommendations [18,19].

The cultivation of cover crops, although extensively studied for improving soil quality, is a large subject because the answers to various problems depend on the cultivated species, main crop, soil type, climatic conditions, N supply for the main crop, and so on. Another gap that needs to be studied is the interaction between soil management and fertilization in soils in tropical regions, and its relations on the N availability to plants, as it has a direct impact on N fertilization requirement. Additionally, long-term studies evaluating this effect in tropical climate soils are scarce.

Measuring the impacts of soil management practices on N availability is essential for providing more accurate fertilization recommendations, which account for the soil's capacity to supply N and result in the more rational use of fertilizers, as well as economic, environmental, and agronomic improvements [19]. Considering the effects of N fertilizer on both plant- and soil-related processes, the combination of these two factors must be

studied, especially in long-term trials, to identify the best management strategies that increase SOC and N availability. Chemical methods to assess N availability have already been tested in soils in the northwest region of the state of São Paulo, Brazil [14,20], showing a strong correlation with dry matter accumulation and N uptake by plants. However, studies involving the soil N fractionation or water-soluble forms of C or N in response to management and nitrogen fertilization are still scarce but are important for measuring the impact of such systems on soil N availability. Thus, our hypotheses in this study were: the cultivation of legumes and nitrogen fertilization increases the total and soluble contents of C and N in soil, in addition to N forms such as amino acids and amino sugars, and, consequently, the potentially available N; the cultivation of grasses results in the same increase of the soil contents of C and N, but only in the presence of mineral N fertilizer. Thus, our objectives were to evaluate the effects of cover crops species and N fertilizer doses applied in top dressing on the SOC, STN, WEOC, and WEON on soil organic N forms and on potentially available N in a long-term experiment with maize cultivation in succession to cover crops.

## 2. Materials and Methods

### 2.1. Study Area

The experiment was conducted in a typic Eutrustox (Brazilian classification: Latossolo vermelho distroférrico) [21,22], located on the experimental farm of the College of Agricultural and Veterinary Sciences (FCAV-Unesp) in Jaboticabal, São Paulo, Brazil, 21°15′22″ S and 48°16′43″ W. The soil topography was gently sloping, the average altitude was 595 m, and the climate was Cwa (humid subtropical, with a dry winter) according to the Koppen International Classification System. The average annual precipitation was 1424 mm, which was concentrated in the months of October to May; the average annual temperature was 22.2 °C.

### 2.2. Soil Analyses

The chemical characterization and granulometry of the soil in samples collected before the experiment at soil depths of 0–20, 20–40, and 40–60 cm are presented in Table 1. The methods used for chemical analysis and the determination of particle size were described in Raij et al. [23] and Camargo et al. [24], respectively.

**Table 1.** Soil chemical attributes and granulometry of experimental area before the experiment combining cover crops and nitrogen doses.

| Depth [†] | Ph [‡] CaCl$_2$ | SOC | Resin P | K$^+$ | Ca$^{2+}$ | Mg$^{2+}$ | H + Al | CEC | BS | Clay | Silt | Sand |
|---|---|---|---|---|---|---|---|---|---|---|---|---|
| cm | | g kg$^{-1}$ | mg kg$^{-1}$ | | | mmolc kg$^{-1}$ | | | | % | | g kg$^{-1}$ | |
| 0–20 | 4.5 | 13.9 | 70 | 3.7 | 16 | 7 | 56 | 83 | 32 | 474 | 27 | 499 |
| 20–40 | 4.3 | 10.4 | 14 | 2.4 | 12 | 5 | 50 | 70 | 28 | 497 | 22 | 481 |
| 40–60 | 4.5 | 9.0 | 8 | 1.4 | 11 | 5 | 41 | 58 | 29 | 504 | 29 | 467 |

[†] Soil sampling depth. [‡] pH: soil pH measured in 0.01 mol L$^{-1}$ CaCl$_2$ solution; SOC: soil organic carbon; resin P: available P extracted with anion exchange resin K, Ca, and Mg: exchangeable K, Ca, and Mg, respectively; H + Al: total acidity; CEC: cation exchange capacity; BS: base saturation; clay, silt, and sand: total contents of clay, silt, and sand, respectively.

### 2.3. Experiment, Cultural Practices, and Management

The experiment was set up in the second half of 2000 after the conventional tillage system stopped; the land was then cultivated under a no-till system, which has been maintained ever since, with cover crop cultivation prior to maize sowing. In all years, cover crops were sown in September or October and chemically desiccated in December or January, just before the maize was sown. This management practice was repeated every year until the 2012/2013 season, which is the season evaluated in this study.

The experimental design was random blocks with split plots and four repetitions. The five main treatments in the plots were the cover crops (rattlepod (*Crotalaria juncea*), jack bean (*Canavalia ensiformis*), lablab bean (*Dolichos lablab*), millet (*Pennisetum americanum*), velvet bean (*Mucuna cinerea*)) and fallow. The split-plots (secondary treatments) were one of two doses of N fertilizer (urea) applied as topdressing fertilization in the maize crop (0 and 120 kg N ha$^{-1}$). The plots dimensions were $18 \times 7$ m (126 m$^2$), and the subplots were $4.5 \times 7$ m (31.5 m$^2$).

*2.4. Collection of Samples and Evaluated Attributes*

Soil and plant samples were collected in the 2012/2013 growing season, which usually starts in October. The analyses were conducted after 12 years of continuous cultivation under no tillage with the same cover crops species. In this season, the cover crops were sown on 19 October, and no fertilization was performed. Approximately 60 days after cover crop seedling emergence, the cover crops shoots were sampled by collecting all biomass from 1 m$^2$ in three random spots. The cover crops samples were oven-dried to determine the dry matter yield (DM). After drying, the plant material was ground, which then used to determine the contents of neutral detergent fiber, acid detergent fiber, and lignin using a sequential extraction system Ankon Fiber Analyzer (Ankon Technology Corporation, Fairport, NY, USA). The results were used to calculate the contents of hemicellulose, cellulose, and lignin [25]. The ground material was later used to determine the total contents of C and N using the dry combustion method with an Elemental Analyzer LECO® CN 628 (Leco corp., St Joseph, MI, USA).

On 7 January, the cover crops were chemically desiccated, and the maize was sown the next day. Hybrid seeds with a 100% germination rate (cv. BM 840 pro, Biomatrix, Patos de Minas, Brazil) were used, with 0.9 m spacing between crop rows and 5 seeds per meter of row (55,500 plants ha$^{-1}$). Fertilization during sowing consisted of 28 kg N ha$^{-1}$, 98 kg P$_2$O$_5$ ha$^{-1}$, and 56 kg K$_2$O ha$^{-1}$ in the seed furrow. The N topdressing fertilization was performed 40 days after maize seedling emergence (V8 stage), using urea applied to the soil surface in the crop rows. Soil samples were collected to determine C and N forms, organic N fractions, and potentially available N using chemical extraction methods 30 days after desiccation. Soil samples were collected with an auger from the 0–5, 5–10, 10–20, and 20–40 cm depths, avoiding the crop rows. Each sample was obtained by combining the 20 subsamples collected in the plot area. The samples were then air-dried, crushed, sieved (open mesh < 2.0 mm), and stored at room temperature (25 °C).

Soil organic C (SOC) and soil total N (STN) were measured according to the methods described by Raij et al. [23]. The soluble organic forms of C (WEOC) and N (WEON) were extracted using deionized water heated to 70 °C in a water bath for 18 h [26]. The WEOC was determined using the K$_2$Cr$_2$O$_7$ wet digestion method, and WEON by Kjeldahl digestion and subtraction of the mineral N contents in the extract.

The soil N fractionation was performed according to Mulvaney et al. [16], in which soil N is hydrolyzed in an acidic solution. Soil hydrolysates were obtained by treating 5 g soil samples with 20 mL of 6 mol L$^{-1}$ HCl and two drops of octyl alcohol in a 10-mL digestion tube stoppered with a glass funnel and heated at 115 °C for 12 h. After heating, the suspension was filtered using Whatman no. 51 filter paper, and its acidity neutralized by the addition of NaOH solution [17]. The N forms were fractionated using diffusion techniques [27] to determine total hydrolyzable N, hydrolyzable NH$_4^+$, NH$_4^+$ + amino sugar, and amino acid N contents. The diffused N was quantified by titration using standardized H$_2$SO$_4$ solution (~0.01 mol L$^{-1}$) with an Automatic Titrator Titrino plus 848 (Metrohn, Herisau, Switzerland). The amino sugar N fraction (AS-N) content was calculated as "NH$_4^+$ + amino sugar"–NH$_4^+$, and hydrolyzable unidentified N (HU-N) as total hydrolyzable N–"NH$_4^+$ + amino sugar"–amino acid N. The fractions hydrolyzable ammonium N (NA-N) and amino acid N (AA-N) were directly determined, and acid-insoluble N (AI-N) was determined as the Kjeldahl total N in the soil residue from acid hydrolysis.

Potentially available N was determined by extraction with hot KCl solution (KCl-N) and direct steam distillation (DSD-N). The KCl-N was extracted with 2 mol L$^{-1}$ KCl solution heated to 100 °C for 4 h [28], and DSD-N was extracted with 10 mol L$^{-1}$ NaOH solution using a nitrogen steam distiller [29]. The N content in both methods was determined by titration.

*2.5. Data Analyses*

The data was evaluated for normality of residues using the Shapiro–Wilk test, and for homoscedasticity of variances using the Levene's test. Soil total and water-soluble C and N contents, N fractions, and potentially available N were submitted to analysis of variance (ANOVA), considering a factorial scheme of $6 \times 2$ (6 cover crops as main treatments, and 2 nitrogen doses as secondary treatments), evaluating the individual effects of each factor and their interactions. Whenever the treatments effects were significant, the means were compared using Tukey's test ($p < 0.05$). The data from all soil depths were also analyzed by a correlation analysis to explore the relationships among the studied parameters.

**3. Results**

*3.1. Soil Total and Water-Extractable Organic C and N Contents*

The compositions of the cover crops differed (Table 2), and thus the return of biomass-N to the soil ranged from 74 to 245 kg N ha$^{-1}$. In most of cases, N recommendations for maize are around 200 kg ha$^{-1}$ according to the recommendations for this region.

**Table 2.** Dry matter yield and composition of cover crops species used in experiment. Results from the 2012/2013 growing season (mean ± standard deviation).

| Cover Crops [a] | DM [b] | TOC | TN | Hemi | Cel | Lig | C:N | Lig:N | Biomass C | Biomass N |
|---|---|---|---|---|---|---|---|---|---|---|
| | t ha$^{-1}$ | | | g kg$^{-1}$ | | | | | T ha$^{-1}$ | kg ha$^{-1}$ |
| Fallow | 4.6 ± 0.6 | 444 ± 26 | 16 ± 3 | 310 ± 35 | 310 ± 37 | 50 ± 9 | 28 ± 5 | 3 ± 0 | 2.04 ± 0.4 | 74 ± 12 |
| Rattlepod | 4.8 ± 0.9 | 410 ± 25 | 34 ± 2 | 144 ± 16 | 186 ± 21 | 46 ± 5 | 12 ± 3 | 1 ± 0 | 1.97 ± 0.3 | 163 ± 31 |
| Jack Bean | 6.9 ± 0.5 | 463 ± 29 | 32 ± 2 | 230 ± 25 | 250 ± 31 | 70 ± 8 | 14 ± 4 | 2 ± 0 | 3.19 ± 0.6 | 221 ± 36 |
| Lablab | 5.1 ± 0.8 | 456 ± 27 | 29 ± 1 | 200 ± 21 | 270 ± 33 | 70 ± 10 | 16 ± 4 | 2 ± 0 | 2.33 ± 0.4 | 148 ± 26 |
| Velvet bean | 5.0 ± 0.5 | 474 ± 28 | 45 ± 3 | 150 ± 16 | 250 ± 31 | 110 ± 12 | 11 ± 2 | 2 ± 0 | 2.37 ± 0.4 | 225 ± 40 |
| Millet | 22.3 ± 2.9 | 456 ± 27 | 11 ± 1 | 270 ± 29 | 330 ± 39 | 50 ± 6 | 41 ± 6 | 4 ± 0 | 10.17 ± 1.8 | 245 ± 44 |

[a] Common names of cover crops species used: fallow: spontaneous vegetation; rattlepod: *Crotalaria juncea*, jack bean: *Canavalia ensiformis*; lablab bean: *Dolichos lablab*; velvet bean: *Mucuna cinerea* and Millet: *Pennisetum americanum*. [b] DM: total aboveground biomass dry matter; TOC: total organic carbon, TN: total nitrogen; Hemi, Cel, and Lig: hemicellulose, cellulose, and lignin contents, respectively; C:N and lig:N: carbon:nitrogen and lignin:nitrogen ratios, respectively; biomass N: total N content in the cover crop's aboveground biomass.

Significant interaction between cover crop species and nitrogen fertilization were observed in SOC and STN contents, but some responses were inverse to what was initially expected, e.g., a decrease in STN content with N application was observed (Table 3). In the 0–5 cm soil layer, in the absence of N fertilizer, velvet bean cover crop promoted the highest SOC and STN contents, at 18.8 and 1.32 g kg$^{-1}$, respectively (Table 3); fallow resulted in the lowest values. Millet cover crop did not increase SOC content, not even in the presence of N fertilizer, but this combination resulted in the high STN content compared with the other treatments (Table 3).

Overall, N fertilizer did not change SOC levels; it only promoted a small increase (12%, on average) in the jack bean and lablab plots (Table 3). Nitrogen fertilizer application, on the other hand, resulted in a decrease in STN content in the velvet bean plots and an increase in that of the millet plots. Jack bean and lablab plots, in general, showed increased SOC contents in the deeper soil layers compared with those of the other cover crops, whereas N fertilizer application had little impact on this attribute at these soil depths. In the 5–10 cm soil layer, jack bean and lablab cover crops resulted in higher STN contents, but in the deeper layers, a decrease in both levels and treatments differences was observed (Table 3).

**Table 3.** Contents of soil organic carbon and nitrogen in samples collected from 0–5, 5–10, 10–20, and 20–40 cm soil layers as a function of cover crops grown before maize and nitrogen fertilizer doses.

| Cover Crops [a] | SOC [b] (g kg$^{-1}$) | | | STN (g kg$^{-1}$) | | |
|---|---|---|---|---|---|---|
| | N0 | N1 | Mean [c] | N0 | N1 | Mean |
| | | | 0–5 cm | | | |
| Fallow | 15.5 Ca [c] | 15.2 Da | 15.4 | 1.09 Ca | 1.07 Ba | 1.08 |
| Rattlepod | 17.8 ABa | 17.0 BCa | 17.4 | 1.20 ABCa | 1.15 ABa | 1.18 |
| Jack Bean | 17.2 ABb | 18.5 ABa | 17.9 | 1.26 Aba | 1.23 Aa | 1.25 |
| Lablab | 15.2 Cb | 17.8 BCa | 16.5 | 1.22 Aba | 1.24 Aa | 1.23 |
| Velvet bean | 18.8 Aa | 19.5 Aa | 19.2 | 1.32 Aa | 1.16 ABb | 1.24 |
| Millet | 16.8 BCa | 16.5 CDa | 16.7 | 1.17 BCb | 1.25 Aa | 1.21 |
| Mean | 16.9 | 17.4 | | 1.21 | 1.18 | |
| | | | 5–10 cm | | | |
| Fallow | 11.2 | 11.5 | 11.4 B | 0.76 Da | 0.81 ABa | 0.79 |
| Rattlepod | 12.5 | 13.0 | 12.8 A | 0.88 Aba | 0.84 ABa | 0.86 |
| Jack Bean | 12.5 | 12.8 | 12.6 A | 0.75 Db | 0.86 Aa | 0.81 |
| Lablab | 12.2 | 12.2 | 12.2 AB | 0.93 Aa | 0.83 ABb | 0.88 |
| Velvet bean | 11.5 | 12.0 | 11.8 AB | 0.85 BCa | 0.77 Bb | 0.81 |
| Millet | 11.8 | 12.2 | 12.0 AB | 0.80 CDa | 0.81 ABa | 0.81 |
| Mean | 12.0 | 12.3 | | 0.83 | 0.82 | |
| | | | 10–20 cm | | | |
| Fallow | 10.0 | 10.2 | 10.2 AB | 0.64 | 0.66 | 0.65 AB |
| Rattlepod | 10.5 | 10.5 | 10.5 AB | 0.69 | 0.63 | 0.66 AB |
| Jack Bean | 11.0 | 11.0 | 11.0 A | 0.57 | 0.64 | 0.61 B |
| Lablab | 11.0 | 11.0 | 11.0 A | 0.65 | 0.73 | 0.69 A |
| Velvet bean | 10.5 | 10.0 | 10.2 AB | 0.61 | 0.67 | 0.64 AB |
| Millet | 9.2 | 10.2 | 9.8 B | 0.63 | 0.69 | 0.66 AB |
| Mean | 10.4 | 10.5 | | 0.63 b | 0.67 a | |
| | | | 20–40 cm | | | |
| Fallow | 10.0 Aa | 8.2 Bb | 9.1 | 0.63 a | 0.59 ABa | 0.61 |
| Rattlepod | 9.5 Aab | 9.0 ABa | 9.3 | 0.65 a | 0.58 Bb | 0.62 |
| Jack Bean | 8.5 BCDb | 9.8 Aa | 9.2 | 0.61 b | 0.68 Aa | 0.65 |
| Lablab | 8.2 CDb | 9.8 Aa | 9.0 | 0.59 a | 0.61 ABa | 0.60 |
| Velvet bean | 9.0 ABCa | 9.5 Aa | 9.3 | 0.61 a | 0.60 ABa | 0.61 |
| Millet | 7.8 Da | 8.2 Ba | 8.0 | 0.58 a | 0.60 ABa | 0.59 |
| Mean | 8.8 | 9.1 | | 0.61 | 0.61 | |

[a] Cover crop species (fallow: spontaneous vegetation; rattlepod: Crotalaria juncea, jack bean: Canavalia ensiformis; lablab bean: Dolichos lablab; velvet bean: Mucuna cinerea; millet: Pennisetum americanum) grown before maize under no-till system. N0 and N1: 0 and 120 kg N ha$^{-1}$, respectively, applied in the maize crop as topdressing fertilization. [b] SOC and STN: soil organic carbon and total nitrogen, respectively. [c] Means followed by different letters, uppercase in the lines and lowercase in the column, differ from each other according to Tukey's test ($p < 0.05$).

Up to the 10–20 cm layer, rattlepod resulted in higher WEOC contents in the N fertilized plots, while in the non-fertilized, jack bean and lablab resulted in higher levels in most of soil layers (Table 4). The WEON contents, in the 0–5 and 5–10 cm soil layers, were increased by the cover crops in comparison to fallow (Table 4). Generally, fertilization decreased WEON contents, however it increased in some cases, such as the jack bean plots at the 5–10 cm, and the velvet bean and millet at the 20–40 cm soil layer (Table 4).

**Table 4.** Contents of water-extractable organic carbon and nitrogen in soil samples collected from 0–5, 5–10, 10–20, and 20–40 cm depth as a function of cover crops grown before maize and nitrogen fertilizer doses.

| Cover Crops [a] | WEOC [b] (mg dm$^{-3}$) | | | WEON (mg dm$^{-3}$) | | |
|---|---|---|---|---|---|---|
| | N0 | N1 | Mean [c] | N0 | N1 | Mean |
| *0–5 cm* | | | | | | |
| Fallow | 279 Db [c] | 360 Ca | 320 | 49.0 Da | 53.4 Ca | 51.2 |
| Rattlepod | 398 Bb | 430 Aa | 414 | 64.3 BCa | 68.6 Aa | 66.5 |
| Jack Bean | 370 Ca | 356 Ca | 363 | 73.4 Aa | 60.8 Bb | 67.1 |
| Lablab | 440 Aa | 429 Aa | 435 | 67.7 ABa | 69.1 Aa | 68.4 |
| Velvet bean | 390 BCa | 402 Ba | 396 | 64.7 BCa | 59.9 Bb | 62.3 |
| Millet | 370 Cb | 402 Ba | 386 | 60.8 Ca | 63.0 ABa | 61.9 |
| Mean | 375 | 397 | | 63.3 | 62.5 | |
| *5–10 cm* | | | | | | |
| Fallow | 162 Cb | 186 BCa | 174 | 29.9 BCa | 30.8 Ba | 30.4 |
| Rattlepod | 172 Cb | 239 Aa | 206 | 33.2 ABCa | 34.4 Ba | 33.8 |
| Jack Bean | 218 Aa | 175 CDb | 197 | 36.2 Ab | 41.3 Aa | 38.8 |
| Lablab | 211 ABa | 159 Db | 185 | 28.3 Ca | 31.2 Ba | 29.8 |
| Velvet bean | 179 Cb | 236 Aa | 208 | 35.4 Aba | 34.4 Ba | 34.9 |
| Millet | 188 BCa | 204 Ba | 196 | 38.4 Aa | 31.9 Bb | 35.2 |
| Mean | 188 | 200 | | 33.6 | 34.0 | |
| *10–20 cm* | | | | | | |
| Fallow | 92 Bb | 148 ABa | 120 | 13.9 | 17.2 | 15.5 B |
| Rattlepod | 110 Bb | 156 Aa | 133 | 20.8 | 22.5 | 21.6 A |
| Jack Bean | 169 Aa | 143 ABb | 156 | 22.2 | 21.9 | 22.1 A |
| Lablab | 106 Ba | 116 Ba | 111 | 25.9 | 20.0 | 23.0 A |
| Velvet bean | 110 Bb | 145 ABa | 128 | 21.3 | 20.4 | 20.9 A |
| Millet | 116 Ba | 129 ABa | 123 | 21.8 | 21.7 | 21.8 A |
| Mean | 117 | 140 | | 21.0 | 20.6 | |
| *20–40 cm* | | | | | | |
| Fallow | 88 | 130 | 109 ABC | 15.0 Aa | 13.5 Ca | 14.3 |
| Rattlepod | 94 | 117 | 105 BC | 16.3 Aa | 15.7 BCa | 16.0 |
| Jack Bean | 104 | 124 | 114 AB | 14.9 Aa | 17.3 ABa | 16.1 |
| Lablab | 94 | 114 | 104 BC | 10.6 Bb | 18.9 ABa | 14.8 |
| Velvet bean | 92 | 112 | 102 C | 14.1 ABb | 20.8 Aa | 17.5 |
| Millet | 106 | 132 | 119 A | 12.8 ABb | 16.1 BCa | 14.5 |
| Mean | 96 b | 121 a | | 14.0 | 17.1 | |

[a] Cover crop species (fallow: spontaneous vegetation; rattlepod: Crotalaria juncea, jack bean: Canavalia ensiformis; lablab bean: Dolichos lablab; velvet bean: Mucuna cinerea; millet: Pennisetum americanum) grown before maize, under no-till system. N0 and N1: 0 and 120 kg N ha$^{-1}$, respectively, applied in maize crop as topdressing fertilization. [b] WEOC and WEON: water-extractable organic carbon and nitrogen, respectively. [c] Means followed by different letters, uppercase in lines and lowercase in column, differ from each other according to Tukey's test ($p < 0.05$).

### 3.2. Soil Organic N Fractions

At the 0–5 cm soil layer, N fertilization decreased the HA-N and AS-N fractions by 8 and 22 mg N kg$^{-1}$, respectively, and had no effect in the AA-N fraction (Table 5). Cover crops did not result in changes in these fractions in most of soil layers, but at the 0–5 and 5–10 cm was observed higher contents of AS-N in the lablab and the lowest in the fallow (Table 5). The HA-N and AA-N fractions lowest contents were observed in the millet plots.

**Table 5.** Concentrations of soil nitrogen fractions of hydrolyzable ammonium, amino sugar, and amino acids from soil samples collected from 0–5, 5–10, 10–20, and 20–40 cm depths as a function of cover crops grown before maize and nitrogen fertilizer dose.

| Cover Crops [a] | HA-N [b] (mg kg$^{-1}$) | | | AS-N (mg kg$^{-1}$) | | | AA-N (mg kg$^{-1}$) | | |
|---|---|---|---|---|---|---|---|---|---|
| | N0 | N1 | Mean [c] | N0 | N1 | Mean | N0 | N1 | Mean |
| | | | | 0–5 cm | | | | | |
| Fallow | 36 | 36 | 36 | 241 | 230 | 236 B [c] | 228 | 175 | 202 |
| Rattlepod | 40 | 33 | 36 | 297 | 246 | 271 AB | 208 | 231 | 220 |
| Jack Bean | 31 | 36 | 34 | 325 | 270 | 297 A | 247 | 248 | 248 |
| Lablab | 41 | 25 | 33 | 298 | 280 | 289 A | 247 | 223 | 235 |
| Velvet bean | 43 | 31 | 37 | 283 | 271 | 277 AB | 256 | 227 | 242 |
| Millet | 41 | 26 | 34 | 285 | 296 | 290 A | 245 | 228 | 237 |
| Mean | 39 a | 31 b | | 288 a | 266 b | | 239 | 222 | |
| | | | | 5–10 cm | | | | | |
| Fallow | 33 | 32 | 33 | 164 | 184 | 174 B | 119 | 122 | 121 |
| Rattlepod | 38 | 24 | 31 | 199 | 192 | 196 AB | 139 | 136 | 138 |
| Jack Bean | 25 | 19 | 22 | 205 | 198 | 201 AB | 127 | 152 | 140 |
| Lablab | 24 | 23 | 24 | 197 | 214 | 205 A | 144 | 140 | 142 |
| Velvet bean | 34 | 26 | 30 | 193 | 183 | 188 AB | 136 | 120 | 128 |
| Millet | 37 | 31 | 34 | 204 | 186 | 195 AB | 142 | 116 | 129 |
| Mean | 32 a | 26 b | | 194 | 193 | | 135 | 131 | |
| | | | | 10–20 cm | | | | | |
| Fallow | 24 | 35 | 30 | 171 | 155 | 163 | 87 | 117 | 102 |
| Rattlepod | 37 | 18 | 28 | 171 | 169 | 170 | 94 | 95 | 95 |
| Jack Bean | 23 | 18 | 21 | 183 | 166 | 175 | 118 | 96 | 107 |
| Lablab | 28 | 25 | 27 | 180 | 158 | 169 | 108 | 79 | 94 |
| Velvet bean | 25 | 16 | 21 | 167 | 154 | 161 | 77 | 92 | 85 |
| Millet | 33 | 23 | 28 | 170 | 187 | 179 | 87 | 117 | 102 |
| Mean | 28 a | 22 b | | 174 | 165 | | 95 | 99 | |
| | | | | 20–40 cm | | | | | |
| Fallow | 17 ABa | 8 Bb | 13 | 162 | 194 | 178 | 84 | 96 | 90 AB |
| Rattlepod | 11 Ba | 14 ABa | 13 | 166 | 206 | 186 | 94 | 99 | 96 AB |
| Jack Bean | 13 ABa | 10 ABa | 12 | 171 | 189 | 180 | 109 | 80 | 99 A |
| Lablab | 17 ABa | 19 Aa | 18 | 182 | 194 | 188 | 109 | 80 | 94 AB |
| Velvet bean | 21 Aa | 13 ABb | 17 | 155 | 197 | 176 | 75 | 86 | 81 AB |
| Millet | 11 Ba | 12 ABa | 12 | 169 | 208 | 189 | 72 | 82 | 77 B |
| Mean | 15 | 13 | | 167 b | 198 a | | 91 | 87 | |

[a] Cover crop species (fallow: spontaneous vegetation; rattlepod: *Crotalaria juncea*, jack bean: *Canavalia ensiformis*; lablab bean: *Dolichos lablab*; velvet bean: *Mucuna cinerea*; millet: *Pennisetum americanum*) grown before maize under no-till system. N0 and N1: 0 and 120 kg N ha$^{-1}$, respectively, applied to maize crop as topdressing fertilization. [b] HA-N, AS-N and AA-N: soil nitrogen fractions hydrolyzable ammonium, amino sugar, and amino acids, respectively. [c] Means followed by different letters, uppercase in the lines and lowercase in the column, differ from each other according to Tukey's test ($p < 0.05$).

At the 0–5 cm soil layer, velvet bean resulted in great increase of AI-N fraction, with a mean content of 273 mg kg$^{-1}$, 107 mg N kg$^{-1}$ more than fallow and 45 mg N kg$^{-1}$ more than jack bean, the lowest and second highest content, respectively (Table 6). At this soil layer, N fertilization decrease the mean contents of AI-N from 233 to 202 mg kg$^{-1}$. Nitrogen fertilization also decreased AI-N contents at 5–10 cm soil layer. In general, there was no effect of cover crops on HU-N and AI-N contents, but rattlepod and millet increased the HU-N at the 5–10 and 20–40 cm soil layers, respectively (Table 6).

**Table 6.** Concentrations of hydrolyzable unidentified and acid-insoluble soil nitrogen fractions from soil samples collected from 0–5, 5–10, 10–20, and 20–40 cm depths as a function of cover crops grown before maize and nitrogen fertilizer dose.

| Cover Crops [a] | HU-N [b] (mg kg$^{-1}$) | | | AI-N (mg kg$^{-1}$) | | |
|---|---|---|---|---|---|---|
| | N0 | N1 | Mean [c] | N0 | N1 | Mean |
| 0–5 cm | | | | | | |
| Fallow | 92 | 182 | 137 | 171 | 162 | 166 B [c] |
| Rattlepod | 163 | 114 | 139 | 233 | 195 | 214 AB |
| Jack Bean | 123 | 113 | 118 | 227 | 228 | 228 AB |
| Lablab | 85 | 76 | 81 | 237 | 196 | 216 AB |
| Velvet bean | 120 | 124 | 122 | 296 | 250 | 273 A |
| Millet | 66 | 85 | 76 | 237 | 184 | 210 B |
| Mean | 108 | 116 | | 233 a | 202 b | |
| 5–10 cm | | | | | | |
| Fallow | 50 | 44 | 47 AB | 142 | 105 | 124 |
| Rattlepod | 94 | 112 | 103 A | 140 | 114 | 127 |
| Jack Bean | 43 | 77 | 60 AB | 143 | 128 | 136 |
| Lablab | 49 | 33 | 41 B | 205 | 142 | 174 |
| Velvet bean | 67 | 75 | 71 AB | 140 | 134 | 137 |
| Millet | 39 | 86 | 62 AB | 151 | 144 | 148 |
| Mean | 57 | 71 | | 153 a | 128 b | |
| 10–20 cm | | | | | | |
| Fallow | 62 | 70 | 66 | 122 | 86 | 104 |
| Rattlepod | 50 | 49 | 50 | 111 | 120 | 116 |
| Jack Bean | 39 | 71 | 55 | 123 | 103 | 113 |
| Lablab | 78 | 68 | 73 | 157 | 135 | 146 |
| Velvet bean | 82 | 71 | 77 | 120 | 133 | 127 |
| Millet | 74 | 97 | 86 | 127 | 119 | 123 |
| Mean | 64 | 71 | | 127 | 116 | |
| 20–40 cm | | | | | | |
| Fallow | 73 | 73 | 73 AB | 98 | 109 | 104 |
| Rattlepod | 69 | 28 | 49 B | 96 | 106 | 101 |
| Jack Bean | 48 | 69 | 58 AB | 90 | 137 | 114 |
| Lablab | 48 | 54 | 51 B | 124 | 126 | 125 |
| Velvet bean | 80 | 73 | 76 AB | 111 | 106 | 109 |
| Millet | 93 | 71 | 82 A | 104 | 98 | 101 |
| Mean | 69 | 61 | | 104 | 114 | |

[a] Cover crop species (fallow: spontaneous vegetation; rattlepod: *Crotalaria juncea*, jack bean: *Canavalia ensiformis*; lablab bean: *Dolichos lablab*; velvet bean: *Mucuna cinerea*; millet: *Pennisetum americanum*) grown before maize under no-till system. N0 and N1: 0 and 120 kg N ha$^{-1}$, respectively, applied to maize crop as topdressing fertilization. [b] HU-N, AS-N, and AI-N: soil nitrogen fractions, hydrolyzable unidentified, and acid insoluble, respectively. [c] Means followed by different letters, uppercase in the lines and lowercase in the column, differ from each other according to Tukey's test ($p < 0.05$).

### 3.3. Potentially Available Nitrogen and Correlations

In respect to N fertilization, in the 0–5 cm soil layer, the potentially available N analyses differed between the methods: with DSD-N, it increased; with KCl-N, it decreased with fertilization (Table 7). There was no agreement in respect to cover crops, although both methods showed that the fallow plots had the lowest N availability; the behavior of the methods differed depending on the cover crop: the highest DSD-N levels were observed in velvet bean plots and the highest KCl-N levels in lablab plots (Table 7). In the 5–10 cm soil layer, the fertilization effects were similar to those in the 0–5 cm layer, and the DSD-N levels were highest in the velvet bean plots.

**Table 7.** Soil potentially available nitrogen measured by direct steam distillation and hot KCl methods in soil samples collected from 0–5, 5–10, 10–20, and 20–40 cm layers as a function of cover crops grown before maize and nitrogen fertilizer doses.

| Cover Crops [a] | DSD-N [b] (mg kg$^{-1}$) | | | KCl-N (mg kg$^{-1}$) | | |
|---|---|---|---|---|---|---|
| | N0 | N1 | Mean [c] | N0 | N1 | Mean |
| *0–5 cm* | | | | | | |
| Fallow | 102 | 103 | 103 B | 21.1 | 16.6 | 18.8 C [c] |
| Rattlepod | 113 | 117 | 115 A | 25.5 | 21.6 | 23.6 B |
| Jack Bean | 113 | 127 | 120 A | 27.9 | 24.3 | 26.1 A |
| Lablab | 116 | 116 | 116 AB | 29.3 | 23.4 | 26.4 A |
| Velvet bean | 119 | 135 | 127 A | 24.9 | 22.4 | 23.7 B |
| Millet | 112 | 117 | 114 AB | 24.6 | 20.1 | 22.3 B |
| Mean | 112 b | 119 a | | 25.6 a | 21.4 b | |
| *5–10 cm* | | | | | | |
| Fallow | 83 | 90 | 86 B | 19.2 | 14.2 | 16.7 |
| Rattlepod | 89 | 95 | 92 AB | 19.0 | 17.8 | 18.4 |
| Jack Bean | 93 | 96 | 94 AB | 19.8 | 19.4 | 19.6 |
| Lablab | 95 | 96 | 95 AB | 19.8 | 16.8 | 18.3 |
| Velvet bean | 102 | 98 | 100 A | 20.8 | 18.2 | 19.5 |
| Millet | 89 | 103 | 96 AB | 18.1 | 15.8 | 17.0 |
| Mean | 92 b | 96 a | | 19.5 a | 17.0 b | |
| *10–20 cm* | | | | | | |
| Fallow | 29 | 34 | 31 C | 18.0 Aa | 12.8 Bb | 15.4 |
| Rattlepod | 53 | 44 | 48 AB | 14.8 ABa | 12.6 Ba | 13.7 |
| Jack Bean | 45 | 64 | 54 A | 13.8 Bb | 17.3 Aa | 15.6 |
| Lablab | 55 | 37 | 46 ABC | 13.0 Ba | 12.8 Ba | 12.9 |
| Velvet bean | 51 | 46 | 49 AB | 17.1 Aa | 13.7 Bb | 15.4 |
| Millet | 42 | 32 | 37 BC | 12.6 Ba | 11.8 Ba | 12.2 |
| Mean | 46 | 43 | | 14.9 | 13.5 | |
| *20–40 cm* | | | | | | |
| Fallow | 49 | 40 AB | 45 | 14.7 ABa | 12.0 Ab | 13.4 |
| Rattlepod | 34 | 54 A | 44 | 13.3 ABCa | 11.7 ABa | 12.5 |
| Jack Bean | 45 | 51 A | 48 | 11.8 BCDa | 10.4 ABa | 11.1 |
| Lablab | 49 | 24 B | 37 | 9.3 Da | 8.8 Ba | 9.1 |
| Velvet bean | 34 | 40 AB | 37 | 14.9 Aa | 11.7 ABb | 13.3 |
| Millet | 48 | 34 AB | 41 | 11.4 CDa | 10.8 ABa | 11.1 |
| Mean | 43 | 41 | | 12.6 | 10.9 | |

[a] Cover crop species (fallow: spontaneous vegetation; rattlepod: *Crotalaria juncea*, jack bean: *Canavalia ensiformis*; lablab bean: *Dolichos lablab*; velvet bean: *Mucuna cinerea*; millet: *Pennisetum americanum*) grown before maize, under no-till system. N0 and N1: 0 and 120 kg N ha$^{-1}$, respectively, applied in maize crop as topdressing fertilization. [b] DSD-N and KCl-N: soil potentially available N extracted by direct steam distillation and hot KCl solution, respectively. [c] Means followed by different letters, uppercase in the lines and lowercase in the column, differ from each other according to Tukey's test ($p < 0.05$).

Almost all correlations were significant, and the correlation coefficients are presented in Table 8. The WEOC and WEON very strongly correlated with SOC, 0.94 * and 0.95 *, respectively; and with STN, 0.96 * and 0.97 *, respectively (Figure 1A–D, and Table 8). The correlation between STN and soil N fractions was very strong (r > 0.80), except for with HA-N and HU-N (0.67 * and 0.59 *, respectively) (Figure 2A,D and Table 8). WEOC and WEON behaved similarly, being strongly correlated with most of the N fractions, DSD-N and KCl-N (Figure 3A–D, and Table 8). The measurements, DSD-N and KCl-N, produced similar results in most of the correlations studied (Table 8). However, KCl-N had slightly stronger correlations with HA-N and AI-N than DSD-N.

**Table 8.** Correlation coefficients among soil carbon and nitrogen forms, nitrogen fractions, and potentially available nitrogen observed in a long-term no-till experiment with different cover crops and nitrogen fertilizer doses.

| | SOC | STN | WEOC | WEON | HA-N | AS-N | AA-N | HU-N | AI-N | KCl-N |
|---|---|---|---|---|---|---|---|---|---|---|
| SOC [a] | 1 | | | | | | | | | |
| STN | 0.96 * | 1 | | | | | | | | |
| WEOC | 0.94 * | 0.95 * | 1 | | | | | | | |
| WEON | 0.96 * | 0.97 * | 0.97 * | 1 | | | | | | |
| HA-N | 0.68 * | 0.67 * | 0.61 * | 0.67 * | 1 | | | | | |
| AS-N | 0.86 * | 0.91 * | 0.90 * | 0.91 * | 0.50 * | 1 | | | | |
| AA-N | 0.94 * | 0.96 * | 0.94 * | 0.96 * | 0.66 * | 0.92 * | 1 | | | |
| HU-N | 0.60 * | 0.59 * | 0.60 * | 0.58 * | 0.39 * | 0.51 * | 0.51 * | 1 | | |
| AI-N | 0.89 * | 0.90 * | 0.86 * | 0.87 * | 0.64 * | 0.85 * | 0.88 * | 0.51 * | 1 | |
| KCl-N | 0.88 * | 0.86 * | 0.83 * | 0.88 * | 0.69 * | 0.78 * | 0.86 * | 0.45 * | 0.82 * | 1 |
| DSD-N | 0.89 * | 0.89 * | 0.87 * | 0.89 * | 0.65 * | 0.77 * | 0.87 * | 0.45 * | 0.78 * | 0.85 * |

[a] SOC: soil total organic carbon; STN: soil total nitrogen; WEOC and WEON: water-extractable organic carbon and nitrogen, respectively; HA-N, AS-N, AA-N, HU-N, and AI-N: soil nitrogen fractions hydrolyzable ammonium, amino sugar, amino acids, hydrolyzable unidentified, and acid insoluble, respectively; KCl-N and DSD-N: potentially available nitrogen extracted with hot KCl solution and direct steam distillation, respectively. * Significant correlation ($p < 0.05$).

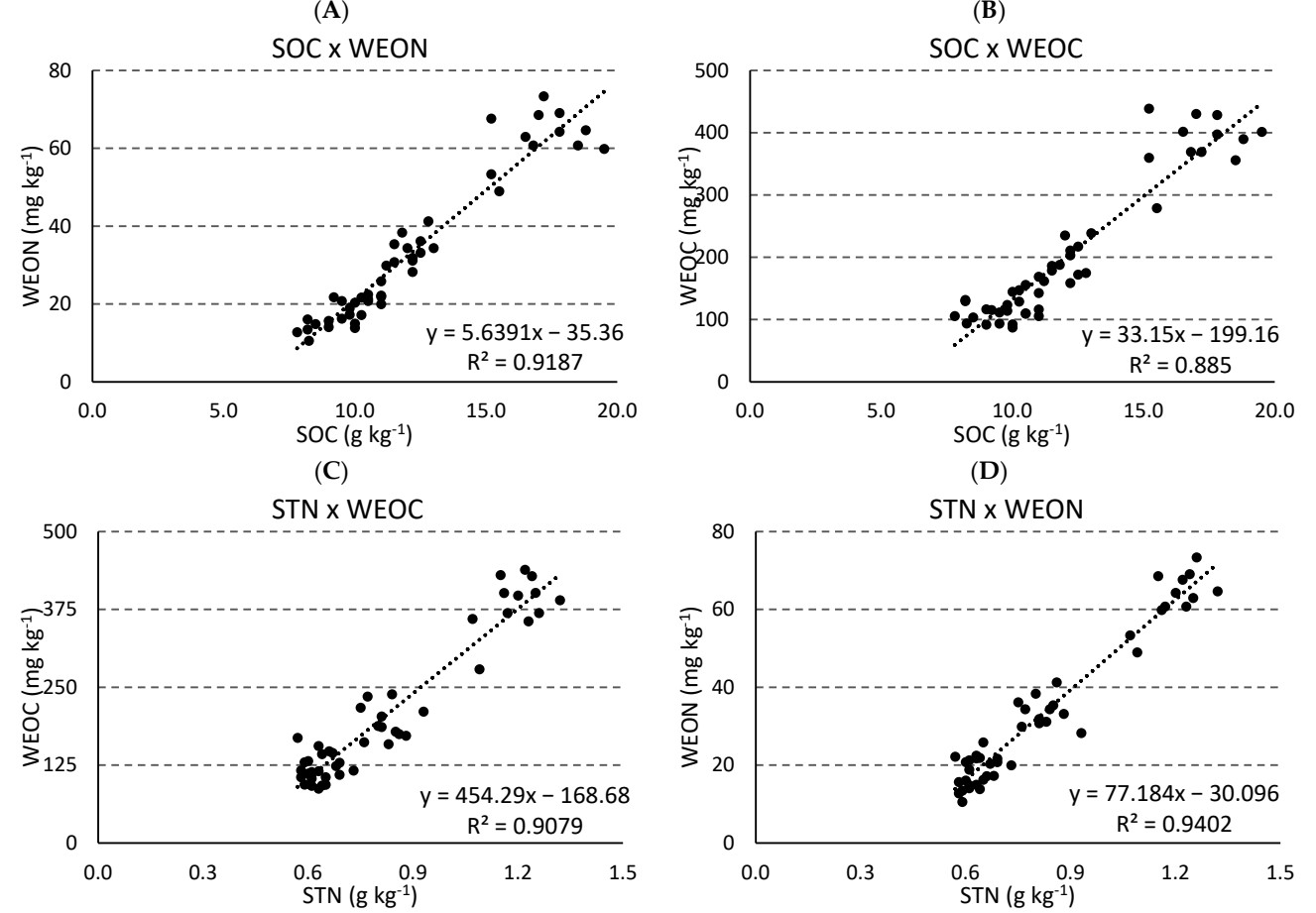

**Figure 1.** Linear relationship between soil organic carbon and water extractable organic carbon (**A**); soil organic carbon and water extractable organic nitrogen (**B**); soil total nitrogen and water extractable organic carbon (**C**); and soil total nitrogen and water extractable organic nitrogen (**D**). Equations and coefficients of determination are displayed in each graph.

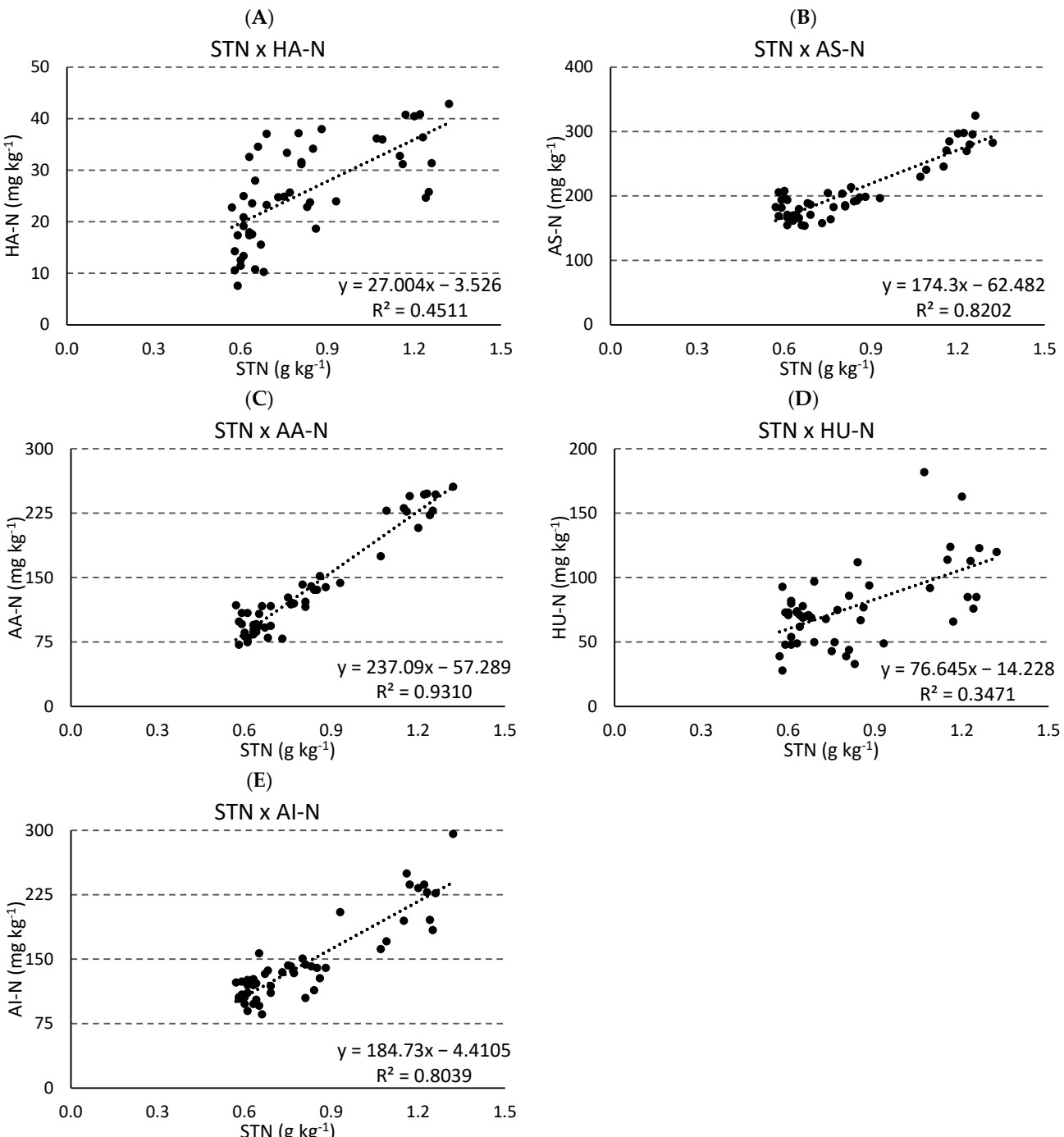

**Figure 2.** Linear relationship between soil total nitrogen and soil nitrogen fractions; hydrolyzable ammonium (**A**), amino sugar (**B**), amino acids (**C**), hydrolyzable unidentified (**D**), and acid insoluble (**E**). Equations and coefficients of determination are displayed in each graph.

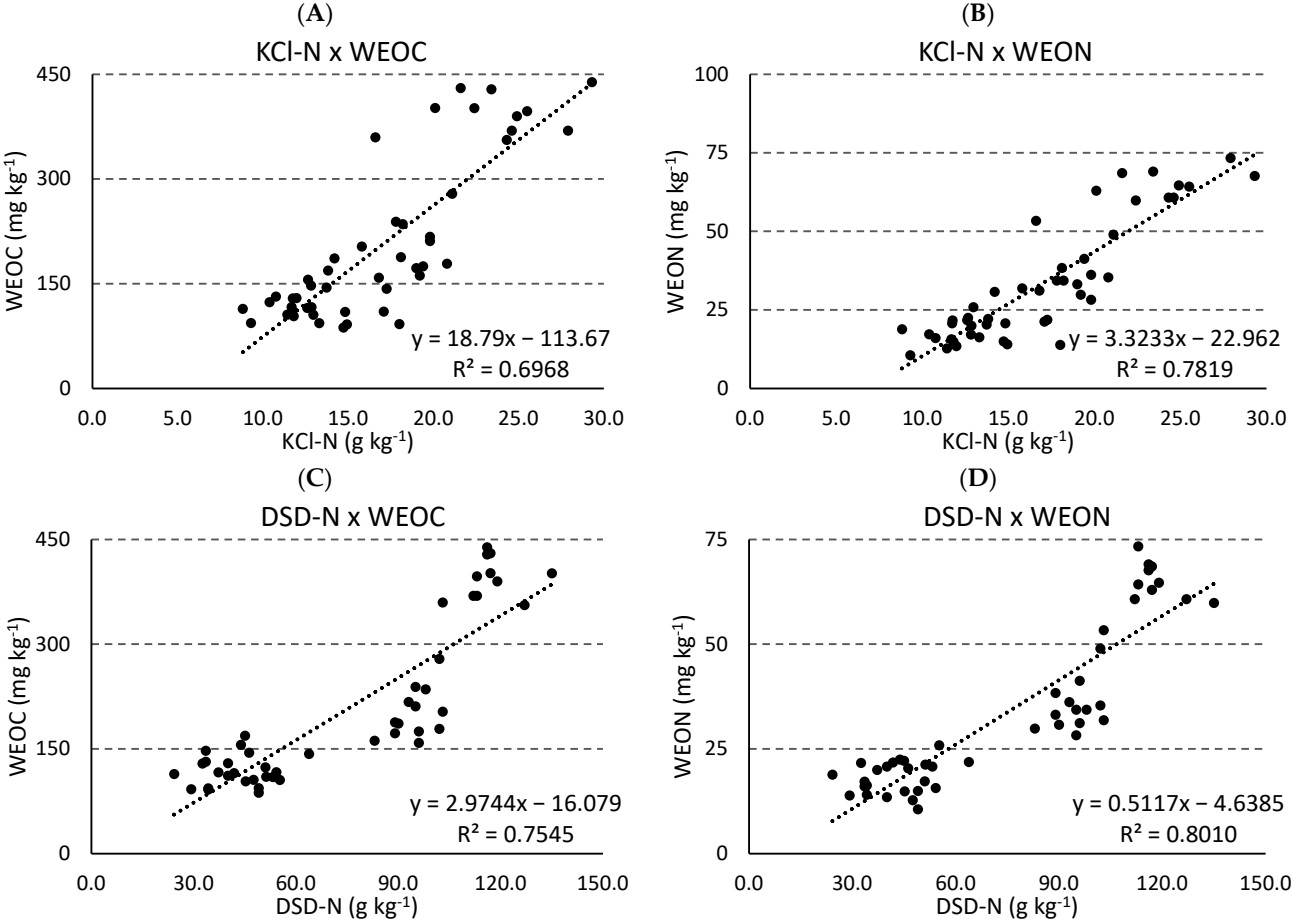

**Figure 3.** Linear relationship between potentially available nitrogen extracted with hot KCl solution and direct steam distillation and water extractable carbon and nitrogen; hot KCl and water extractable organic carbon (**A**); hot KCl and water extractable organic nitrogen (**B**); direct steam distillation and water extractable organic carbon (**C**); and direct steam distillation and water extractable organic nitrogen (**D**). Equations and coefficients of determination are displayed in each graph.

## 4. Discussion

Despite the increased DM accumulation and higher C:N ratio, cover crop millet was not efficient in increasing the SOC content. This was mainly due to its composition; it is rich in hemicellulose and cellulose (Table 2), which serve as the substrate for the degradation of more stable organic compounds [7]. The low lignin content in millet was also a factor that limited the increase in SOC content; because this compound has a complex structure, high molecular weight, and insolubility, it is difficult to mineralize and has a close relationship with the synthesis of humic substances in the soil [7,30,31]. This explains the increase in SOC after cover crop velvet bean, which, despite the low C:N ratio and DM production, was more efficient in increasing the SOC content. The overall increase in SOC content due to N fertilization may have been a result of the higher DM accumulation in both maize and cover crop, which was promoted by the addition of N [4]. The lower SOC contents and less differentiation among treatments in the underlying layers were due to stratification caused by the no-till system and the non-incorporation of plant residues in the soil [5]. Similar results for the other evaluated attributes were observed for the same reasons, as all attributes were related to the soil organic C and N dynamics.

The increase in STN content in the soil occurred in the millet plots receiving the high N treatment most likely due to the immobilization of N by the microorganisms during the mineralization of this crop residue, which has a high C:N ratio [9,32]. Other reasons for these finding were, first, the high amount of N accumulated in its biomass,

which, despite having a low N content, is offset by its DM yield (Table 2), and, second, its efficiency in absorbing the residual mineral N from soil [33]. The opposite effect observed for velvet bean occurred because, in this case, the residue and the soil were already rich in N (Tables 2 and 3), so the addition of N via fertilization may have accelerated the rate of decomposition of the organic material [7,34]. This effect was also observed in the levels of some fractions of organic N, such as HA-N, AS-N, and AI-N (Tables 5 and 6), as it was by researchers in previous studies [2,11]. This agrees with the results reported by Bettiol et al. [2] in the same experiment, but conducted in the 2015/2016 growing season. In that season, SOC and STN levels decreased in the N fertilized plots (with exception of the plots cultivated with millet). The authors also reported lower maize grain yield and N uptake by maize plants in plots cultivated with millet and fallow due to the lower availability of available N.

Increases in soil microbial activity resulted in increased degradation of organic compounds, which are degraded into smaller molecules, such as those present in WEOC, that can be metabolized by soil microorganisms [15]. When N fertilization was applied, an increase in plant biomass production and the return of biomass to the soil was noted, which could be transformed in WEOC as the biomass was decomposed by soil microorganisms (Table 4). This result was especially evident in the millet and fallow plots. The addition of N also tended to boost soil microbial activity, which contributed to increased WEOC production [15,35]. Contrary to what was observed for WEOC, fertilization decreased the WEON contents in the soil of plots cultivated with jack bean and velvet bean, which are plants with higher N content and whose degradation is favored by the addition of N.

The AS-N and AA-N fractions are the main sources of available N in the soil due to their N content and lability [16,17,20]. The increases in AS-N contents in the surface soil layers was due to the input of organic material and N by the cover crops. Fallow, the cover crop with the lowest N levels, resulted in the lowest contents of these fractions in the soil (Tables 2 and 5). The decreases in HA-N and AS-N contents that were observed after N fertilization were opposite to what was observed by Reddy et al. [36], which indicates that fertilization may have decreased the soil indigenous N supply [2,11]. This difference may have been due to the influence of N addition on the decomposition of soil organic residues and its interaction with the composition of these residues (Table 2), which was previously discussed. Only residues with lower N contents and high C:N ratios positively influenced the accumulation of organic N after N fertilization (Table 3). A similar effect was observed in the AS-N content; only in the millet plots was there an increase of this fraction content due to N fertilization (Table 5), whereas in the rattlepod, jack bean lablab, and velvet bean plots, the N fertilization resulted in decrease of AS-N contents.

The HU-N and AI-N fractions are the most stable N fractions of soil, having low availability and contributing little to plant uptake [17]. The HU-N fraction contents were increased by the cultivation of rattlepod and millet in subsurface layers due to the possible contribution of root debris, whose decomposition may have increased the contents in this fraction. The increased accumulation of AI-N in the soil of plots cultivated with velvet bean and jack bean coincided with the increases in STN contents, and probably occurred for the same reasons (Table 3). Nitrogen fertilization contributed to the decrease in the contents of the AI-N fraction, contrary to what was expected, because this is the most stable fraction of soil N and should be the one that was less impacted by the treatments. Even though AI-N has little influence on available N, its decrease highlights the possible reduction in soil N contents caused by fertilization [11].

Differences between the results of potentially available N evaluated by the DSD-N and KCl-N methods were observed, as these methods extract different fractions of organic N. Whereas DSD-N extracts part of the AS-N, KCl-N extracts mineral N and part of the HA-N and AS-N [29,37]. This is shown in Table 8, where the correlation coefficient between KCl-N and HA-N is slightly higher than that with DSD-N. The AS-N fraction decomposes in strong alkali conditions and can be efficiently extracted by NaOH solutions. This fraction is known as one of the main substrates of N mineralization, and its contents strongly

correlate with the potentially mineralizable N ($N_0$) obtained in laboratory incubations and the N uptake by plants [2,12,14,38]. These characteristics make the DSD-N method more suitable for evaluating the potentially available N, having previously demonstrated strong correlations with N absorbed by plants [2,38].

The strong correlation between the potentially mineralizable N measurements with WEOC and WEON indicate that these forms of C and N can be used to assess soil N availability [2,15]. The strong correlation between N-KCl and soil N fractions that are less relevant to N availability suggests that this method is less efficient than strong alkali methods [2,14].

## 5. Conclusions

The interactions between cover crop species and N fertilization are complex and must be carefully evaluated to ensure that a system that provides better accumulation of soil organic carbon and N availability is chosen. Thus, N-rich plants such as velvet bean should be cultivated in low-N-input systems because the addition of N seems to increase the mineralization of organic compounds and impair the soil organic N build-up. The cultivation of millet is more suitable for systems with high mineral N input, as the large input of organic C needs mineral N to ensure satisfactory microbial activity in these systems, which results in an increase in N availability. Due to the high correlation with potentially available N, both WEON and WEOC can be used as tools to assess the availability of N depending on the type of soil management; studies that assess the efficiency and calibration of these methods are still needed. The study also highlights that nitrogen fertilization may increase the availability of soil organic N in some cases, but this may also cause a decrease in some N fractions, which can lead to the depletion of soil N and its capacity to supply N to plants in the long term. So, more studies and monitoring should be conducted to clarify further details.

**Author Contributions:** Conceptualization, A.C.T.B., M.A.M.B. and M.C.P.d.C.; methodology, A.C.T.B., M.A.M.B., M.C.P.d.C. and M.E.F.; software, L.B.B. and R.S.C.; validation, L.B.B., M.C.T. and R.S.C.; formal analysis, A.C.T.B., M.A.M.B., M.C.T.; investigation, A.C.T.B., M.A.M.B., M.C.P.d.C. and M.E.F.; resources, A.C.T.B., M.A.M.B., M.C.P.d.C. and M.E.F.; data curation, L.B.B., M.C.P.d.C. and R.S.C.; writing—original draft preparation, A.C.T.B., L.B.B. and R.S.C.; writing—review and editing, L.B.B. and R.S.C.; visualization, L.B.B. and R.S.C.; supervision, M.C.P.d.C. and M.E.F.; project administration, A.C.T.B., M.A.M.B. and M.C.P.d.C.; funding acquisition, A.C.T.B., M.A.M.B. and M.C.P.d.C. All authors have read and agreed to the published version of the manuscript.

**Funding:** This work was supported by the Coordenação de Aperfeiçoamento de Pessoal de Nível Superior-Brasil (CAPES) Finance Code 001; Fundação Agrisus.

**Data Availability Statement:** The data that support the findings of this study are available from the corresponding author, Braos, L.B., upon reasonable request.

**Acknowledgments:** The authors would like to thank the Teaching and Research Farm (FCAV) staff for helping to conduct the field experiment.

**Conflicts of Interest:** The authors declare no conflict of interest.

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
