# Peer review of "Soil Carbon and Nitrogen Forms and Their Relationship with Nitrogen Availability Affected by Cover Crop Species and Nitrogen Fertilizer Doses"

_nitrogen, doi:10.3390/nitrogen4010007_

Round 1
Reviewer 1 Report
The manuscript with the title “Soil carbon and nitrogen forms and their relation to nitrogen availability affected by cover crop species and nitrogen fertilizer doses” explores the effects of cover crops and N doses on soil total and soluble C and N contents, N-fractions and potentially available N, in long-term no-till experiment.
Because the soil functions and properties can experience a different dynamic in no-tillage system than in conventional system, there are aspects that must be clarified and researched in order to be able to optimize them. In this regard the paper brings an important contribution to the field.
Introduction
General information about no-tillage is too brief (only a few sentences) although is central to the study. Please also mention that no-tillage is adopted more and more worldwide as a solution to prevent soil erosion. Only some brief advantages are given but important aspects are not mentioned. For example, in no-tillage systems the soil tends to become acidic in a few years and needs correction at few years interval, to ensure adequate soil pH for crop growth, and this is an adopted practice in many countries from temperate climate. Is this a practice also in Brazil? Or how much interest is there for no-tillage in Brazil? Both general pros and cons should be briefly mentioned either here or in discussion section. No-tillage also causes a strong vertical gradient in both soil properties and edaphic microflora activity with influence on soil fertility.
Material and Method
Lines 120-123 – botanic name with italic
Line 114-118 – did the soil pH was corrected by application of soil amendments? Usually in no-tillage system the soil becomes acidic in detriment to crop growth, and every few years the pH unusually gets corrected. What maintenance was done to the soil?
Results
All results are presented as tables, perhaps one graph could be used to highlight the most interesting trends observed?
Best regards.
Author Response
-Introduction
We included the some these information in the introduction.
Material and methods
-Lines 120-123 – botanic name with italic
Done
-Line 114-118 – did the soil pH was corrected by application of soil amendments? Usually in no-tillage system the soil becomes acidic in detriment to crop growth, and every few years the pH unusually gets corrected. What maintenance was done to the soil?
In the first year of experiment, soil acidity was corrected by liming. The liming was calculated to increase soil base saturation to 70% and its was followed by plowing and harrowing. After that, soil acidity was corrected in 2017, with superficial liming. But it was after the current samples were collected.
-All results are presented as tables, perhaps one graph could be used to highlight the most interesting trends observed?
We tried to use graphs but the excessive statistical unfoldings made that really difficult.
Thanks for the help! Hope we answered all questions.
Reviewer 2 Report
The subject of the manuscript corresponds to the subject of the journal and it can potentially be published. At the same time, the manuscript must be substantially improved before publication. I offer a major revision. Below I will give some special comments and questions.
Abstract: Remove the value of the correlation coefficient, but add the overall conclusion of the study and its implications for science.
Unfortunately, the Introduction does not form an idea of the problem and is not a prerequisite for the proposed hypotheses. It is necessary not only to indicate generally known facts, but also to point out contradictions and gaps of knowledge in this field of science.
L93: This part of the text should be divided into subsections (for example, Study area; Design of the experiment; Soil analyses; Data analyses; etc.). How many soil samples were analyzed? Judging by the very high correlation coefficients (Table 8), it was few number of samples.
LL101-107: How similar were the soil properties throughout the farm where the experiment was conducted?
Table 1 and others: Could you provide not only the mean values of the indicators, but also, for instance, the standard error of the mean?
Probably some tables can be transformed into graphs and charts.
LL 121-123: Please use italics for Latin plant names.
Unfortunately, the Discussion in many respects only repeats the Results themselves, and does not provide any generalization and explanation. In addition, the hypotheses formulated in the Introduction were not reflected in the Discussion.
(Wang et al., 2019) not numbered in References
Author Response
Abstract
We removed the correlation coeficient value and added the overall study conclusion
Introduction
Done .
L93: This part of the text should be divided into subsections
Done .
How many soil samples were observed?
Included: Ten subsamples were collected between the lines of the culture, which were joined, homogenized and formed a composite sample .
LL101-107:
The experiment was installed in the same soil: a typical Eutrustox (Brazilian classification: Red Distroferric Latosol). The possible differences are related to the different treatments during the several years of the experiment.
Table 1 and others: Could you provide not only the mean values of the indicators, but also, for example, the standard error of the mean?
Thats possible, but we decided to put only de tukeys test letters (when applicable) for aesthetic reasons, the tables would be confusing with so many numbers
probably some tables can be turned into charts and tables
We tried to use graphs, but too many statistical interactions and measures made it too difficult.
LL 121-123: Please use italics for Latin plant names.
Done .
Unfortunately, the Discussion in many respects just repeats the Results themselves and does not provide any generalizations and explanations. Furthermore, the hypotheses formulated in the Introduction were not reflected in the Discussion.
We disagree. In several parts of the discussion, we explain the effects that occurred and relate them to the presented hypothesis. For example: 'The decrease in HA-N and AS-N contents due to nitrogen fertilization was opposite to that observed by Reddy et al. [36] and indicates that fertilization may be observing N availability [2,11]. This difference may be due to the influence of N addition on the composition of soil organic residues and its interaction with the composition of these residues (Table 2)' .
We made some improvements in the discussion (as requested by other reviewers)
(Wang et al., 2019) not numbered in References:
Withdrawn.
Thanks for the help! Hope we answered all questions.
Reviewer 3 Report
Dear Author,
your manuscript reports interesting results and well fits the Journal’s aim.
However, you have to better report the knowledge on this topic (I add some relevant literature here after) but you logically stated your hypothesis. I also applaud the rigorous statistical analysis. The paper is properly organized and could be published after minor revision.
The technical language is poor and as a result, many parts of the paper are difficult to understand.
The English needs corrections in some parts.
I attach a file with some annotated examples.
Legends of tables and figures need amendment to be self-explicative and able to stand alone without having to refer to the main text and be more communicative.
Tables are not clear in showing results.
Sincerely
Abstract is unclear.
Define abbreviations at first mention.
Use abbreviations at first mention and then consistently throughout.
Define organisms at first mention.
Tables and figures must be able to stand alone without having to refer to the main text.
Basic editing for English is required in some places.
The Methods section is inadequate (for example at which crop stage was the N applied?).
It is unclear if the experiment was repeated in space and/or time.
The Journal guidelines are not followed.
Relevant literature to be added:
https://doi.org/10.1002/agj2.21160 is a systematic review on cover crops that should be cited.
https://doi.org/10.2134/agronj2016.06.0330 reported a remarkable model approach that could be useful in your discussion
https://doi.org/10.2134/agronj2000.925915x found that cover crops were effective at lowering residual soil NO3–N concentrations.
https://doi.org/10.1002/agj2.20791 reported interesting results and demonstrated that cover crop management should be considered to adjust the fertilizer N rate and optimize maize productivity.

Author Response
Abstract
Ln 11-12: Changed, same lines
Material and methods
Ln 106: changed, now line 120
Ln 124 “topdressing”:
Ln 128 “Begin”: replanced by starts
Ln 332: “serves”
Other comments
Abstract is unclear.
A:We performed several changes (others reviewers) to improve the abstract
Define abbreviations at first mention.
Use abbreviations at first mention and then consistently throughout.
A:We correct the abbreviations
Define organisms at first mention.
A:Done
Tables and figures must be able to stand alone without having to refer to the main text.
A:checked
Basic editing for English is required in some places.
A:We submitted the mc to English edit service
The Methods section is inadequate (for example at which crop stage was the N applied?).
A:We performed several changes (others reviewers) to improve this section
It is unclear if the experiment was repeated in space and/or time.
A:We tried to make it clear
The Journal guidelines are not followed.
A: We reviewed the guidelines
About the recommended literature we included two of them, unfortunately we could not have access to the others.
Round 2
Reviewer 2 Report
Unfortunately, the results are still very poorly visualized. Tables are not an ideal solution, their "aesthetics" as stated in the authors' answer is doubtful. The main regularities could be reflected in the graphs, and the tables could be placed in the appendix.
Author Response
Thanks again for the help with the manuscript.
We performed english editing of the manuscript
We included graphs from the correlations ( was also a request from the editor)
About the replacement of the tables by graphs, we discussed with the authors this matter and we all agreed that graphs would improve greatly the visualization of the treatmentes effects. However, we decide to defend the presentation of the data in tables because it is better to explore the statistical unfolding of the interactions of the treatments. The tables make it possible to distinguish the effects of secondary treatments within the main ones and vice versa.
-We considered the suggestion with much appreciation, but decided to keep the tables for the reasons above mentioned.